

# Isolated tooth reveals hidden spinosaurid dinosaur diversity in the British Wealden Supergroup (Lower Cretaceous)

Chris T. Barker[1,2], Darren Naish[3] and Neil J. Gostling[1,3]

[1] Institute for Life Sciences, University of Southampton, Southampton, United Kingdom
[2] Faculty of Engineering and Physical Sciences, University of Southampton, Southampton, United Kingdom
[3] School of Biological Sciences, Faculty of Environment and Life Sciences, University of Southampton, Southampton, United Kingdom

Corresponding authors
Chris T. Barker, ctb1g14@soton.ac.uk
Neil J. Gostling,
n.j.gostling@soton.ac.uk

## ABSTRACT

Isolated spinosaurid teeth are relatively well represented in the Lower Cretaceous Wealden Supergroup of southern England, UK. Until recently it was assumed that these teeth were referable to *Baryonyx*, the type species (*B. walkeri*) and specimen of which is from the Barremian Upper Weald Clay Formation of Surrey. British spinosaurid teeth are known from formations that span much of the c. 25 Ma depositional history of the Wealden Supergroup, and recent works suggest that British spinosaurids were more taxonomically diverse than previously thought. On the basis of both arguments, it is appropriate to doubt the hypothesis that isolated teeth from outside the Upper Weald Clay Formation are referable to *Baryonyx*. Here, we use phylogenetic, discriminant and cluster analyses to test whether an isolated spinosaurid tooth (HASMG G369a, consisting of a crown and part of the root) from a non-Weald Clay Formation unit can be referred to *Baryonyx*. HASMG G369a was recovered from an uncertain Lower Cretaceous locality in East Sussex but is probably from a Valanginian exposure of the Hastings Group and among the oldest spinosaurid material known from the UK. Spinosaurid affinities are both quantitatively and qualitatively supported, and HASMG G369a does not associate with *Baryonyx* in any analysis. This supports recent reinterpretations of the diversity of spinosaurid in the Early Cretaceous of Britain, which appears to have been populated by multiple spinosaurid lineages in a manner comparable to coeval Iberian deposits. This work also reviews the British and global records of early spinosaurids (known mainly from dental specimens), and revisits evidence for post-Cenomanian spinosaurid persistence.

# INTRODUCTION

Spinosaurids are an unusual clade of large-bodied tetanuran theropods best known for the multiple lines of evidence indicating specialisation for a semi-aquatic ecology and the associated controversy over their lifestyle (*Amiot et al., 2010a*; *Bertin, 2010*; *Charig & Milner, 1997*; *Fabbri et al., 2022*; *Hassler et al., 2018*; *Holtz, 1998*; *Hone & Holtz Jr, 2021*; *Ibrahim et al., 2020a*; *Sereno et al., 2022*; *Taquet, 1984*). Spinosaurids are currently known

from Cretaceous deposits and possess a wide spatial distribution, with important specimens coming from England, Brazil, northern Africa, the Iberian Peninsula and Southeast Asia. The clade is generally considered to consist of the sister-clades Baryonychinae (anchored on *Baryonyx walkeri* from southern England) and Spinosaurinae (anchored on *Spinosaurus aegyptiacus*, first described from Egypt though since reported from other north African countries) (*Allain et al., 2012*; *Arden et al., 2019*; *Benson, 2010*; *Bertin, 2010*; *Carrano, Benson & Sampson, 2012*; *Charig & Milner, 1997*; *Holtz, Molnar & Currie, 2004*; *Ibrahim et al., 2020a*; *Ibrahim et al., 2014*; *Mateus & Estraviz-López, 2022*; *Rauhut & Pol, 2019*; *Sereno et al., 1998*; *Sereno et al., 2022*; *Stromer, 1915*; *Sues et al., 2002*). However, several recent analyses suggest that support for this dichotomy may not be as robust as usually supposed (*Barker et al., 2021*; *Evers et al., 2015*; *Sales & Schultz, 2017*).

The fossiliferous Lower Cretaceous (late Berriasian–early Aptian) Wealden Supergroup of southern England is a significant location for the clade, notably following the 1983 discovery of the *Baryonyx walkeri* holotype (*Charig & Milner, 1986*; *Charig & Milner, 1997*). The discovery of *B. walkeri*, represented by a partial skeleton, was integral to the reinterpretation of Spinosauridae (*Naish & Martill, 2007*), and resulted in the realisation that isolated teeth known from throughout the succession—traditionally regarded as crocodilian—also pertain to spinosaurids (*Buffetaut, 2007*; *Buffetaut, 2010*; *Fowler, 2007*). Indeed, among the first dinosaur remains to be scientifically illustrated and described are spinosaurid teeth from the English Wealden Supergroup, discovered in or around 1820 and given the binomial name "*Suchosaurus cultridens*" (*Buffetaut, 2010*; *Owen, 1840–1845*). These were misinterpreted as crocodilian for nearly two centuries (one of the longest cases of taxonomic misidentification), and were not correctly identified as spinosaurid until more recently (*Buffetaut, 2007*; *Buffetaut, 2010*). "*Suchosaurus cultridens*" is currently considered a *nomen dubium*, being best interpreted as an indeterminate spinosaurid (*Mateus et al., 2011*; *Salisbury & Naish, 2011*). More recent finds from the Wealden Supergroup succession on the Isle of Wight include the incomplete skeletons of the baryonychine taxa *Ceratosuchops inferodios* and *Riparovenator milnerae* from the Wessex Formation (*Barker et al., 2021*), and the as-yet-unnamed "White Rock" spinosaurid (a possible spinosaurine) from the overlying Vectis Formation (*Barker et al., 2022*).

Spinosaurid skeletal material is rare (*Hone, Xu & Wang, 2010*), but tooth crowns attributed to the group are regularly discovered; numerous isolated specimens have been reported from England (*Charig & Milner, 1997*; *Fowler, 2007*; *Martill & Hutt, 1996*; *Turmine-Juhel et al., 2019*), Spain (*Alonso & Canudo, 2016*; *Isasmendi et al., 2020*; *Ruiz-Omeñaca et al., 2005*), China (*Buffetaut et al., 2008*; *Shu'an, Pei & Daolin, 2022*), Malaysia (*Sone et al., 2015*), Japan (*Hasegawa et al., 2003*; *Katsuhiro & Yoshikazu, 2017*), Thailand (*Buffetaut & Ingavat, 1986*; *Buffetaut et al., 2019*; *Wongko et al., 2019*), Algeria (*Benyoucef et al., 2015*; *Benyoucef et al., 2022*), Cameroon (*Congleton, 1990*), Morocco (*Richter, Mudroch & Buckley, 2013*), Libya (*Le Loeuff et al., 2010*), Niger (*Sereno et al., 1998*), Tunisia (*Benton et al., 2000*; *Bouaziz et al., 1988*) and Brazil (*Lacerda et al., 2023*; *Medeiros, 2006*; *Sales et al., 2017*) (see also *Bertin (2010)* for further references and notes). Putative spinosaurid dental material may also extend the temporal span of the clade, though reported teeth from the Jurassic of France (*Vullo et al., 2014*), Tanzania (*Buffetaut, 2012*) and Niger

(*Serrano-Martínez et al., 2015*; *Serrano-Martínez et al., 2016*), as well as the Late Cretaceous of China (*Hone, Xu & Wang, 2010*) and Patagonia (*Salgado et al., 2009*), likely belong to other archosaur clades (*Hendrickx et al., 2019*; *Soto, Toriño & Perea, 2020*).

Spinosaurid teeth are specialised and distinctive relative to those of other theropods, and possess a list of autapomorphies (*Hendrickx & Mateus, 2014*; *Hendrickx et al., 2019*). These allow them to be differentiated from the teeth of crocodylomorphs and plesiosaurs, two groups with which they have occasionally been confused (*Bertin, 2010*; *Buffetaut, 2010*; *Hone, Xu & Wang, 2010*; *Sánchez-Hernández, Benton & Naish, 2007*; *Sanguino, 2020*; *Soto, Toriño & Perea, 2020*). Key spinosaurid tooth characters, which are likely adaptions towards piscivory, include conidont (cone-shaped) morphology, fluted enamel surfaces, and veined enamel surface texture (*Charig & Milner, 1997*; *Hendrickx et al., 2019*; *McCurry et al., 2019*). Spinosaurid teeth are not homogenous: those conventionally attributed to baryonychines possess minutely denticulated carinae, while those conventionally attributed to spinosaurines are unserrated and weakly recurved (*Barker et al., 2021*; *Carrano, Benson & Sampson, 2012*; *Hendrickx et al., 2019*). Spinosaurid teeth have been important with respect to discussions on the palaeobiology of the clade: they not only provide data on diet, ecology and lifestyle (*Amiot et al., 2009*; *Amiot et al., 2010a*; *Amiot et al., 2010b*; *Buffetaut, Martill & Escuillié, 2004*; *Hassler et al., 2018*; *Hone & Holtz Jr, 2021*) but also physiology (*Heckeberg & Rauhut, 2020*) and—most importantly for the present study—species-level diversity and palaeoenvironmental and stratigraphic distribution (*Alonso & Canudo, 2016*; *Beevor et al., 2021*; *Fanti et al., 2014*; *Ruiz-Omeñaca et al., 2005*; *Sales et al., 2016*).

Those spinosaurid teeth discovered throughout Wealden Supergroup strata were initially assumed to be referable to *Baryonyx* (albeit not necessarily to *B. walkeri*) on the basis of general similarity. *Charig & Milner (1997)* referred isolated crowns from the Wessex, Upper Weald Clay and "Ashdown Sands" formations to cf. *Baryonyx*. Isolated teeth of the NHMUK collections, some previously referred to "*Megalosaurus*" and "*Suchosaurus*", were also referred to *Baryonyx* by *Milner (2003)*. *Buffetaut (2010)* agreed that many of the "*Suchosaurus*" crowns from the Wealden Supergroup could be attributed to *Baryonyx*. More recently, *Turmine-Juhel et al. (2019)* referred incomplete crowns from the Wadhurst Clay Formation to *Baryonyx* sp. Attributing these various Wealden teeth to *Baryonyx* (or cf. *Baryonyx*) was a reasonable proposal in view of knowledge of Wealden spinosaurid diversity at the time but recent finds demonstrate higher diversity across the supergroup (*Barker et al., 2021*; *Barker et al., 2022*). In addition, it should be noted that these fossils come from strata spanning a time frame (∼25 million years) not considered typical for the duration of a genus-level dinosaur taxon (*Naish, 2011*). However, these teeth differ in several ways from the dentition of the *Baryonyx walkeri* holotype and we consider it plausible that they represent additional taxa (*Buffetaut, 2010*; *Naish, 2011*; *Naish & Martill, 2007*).

A collection of archosaur teeth (HASMG G369) accessioned at Hastings Museum and Art Gallery (East Sussex, UK) includes one specimen (HASMG G369a) bearing the conidont appearance and minute denticles typical of baryonychine spinosaurids. An associated note indicates that these teeth were discovered close to the village of Netherfield in West Sussex (Fig. 1), and from the Purbeck Group, a succession that underlies the Wealden Supergroup

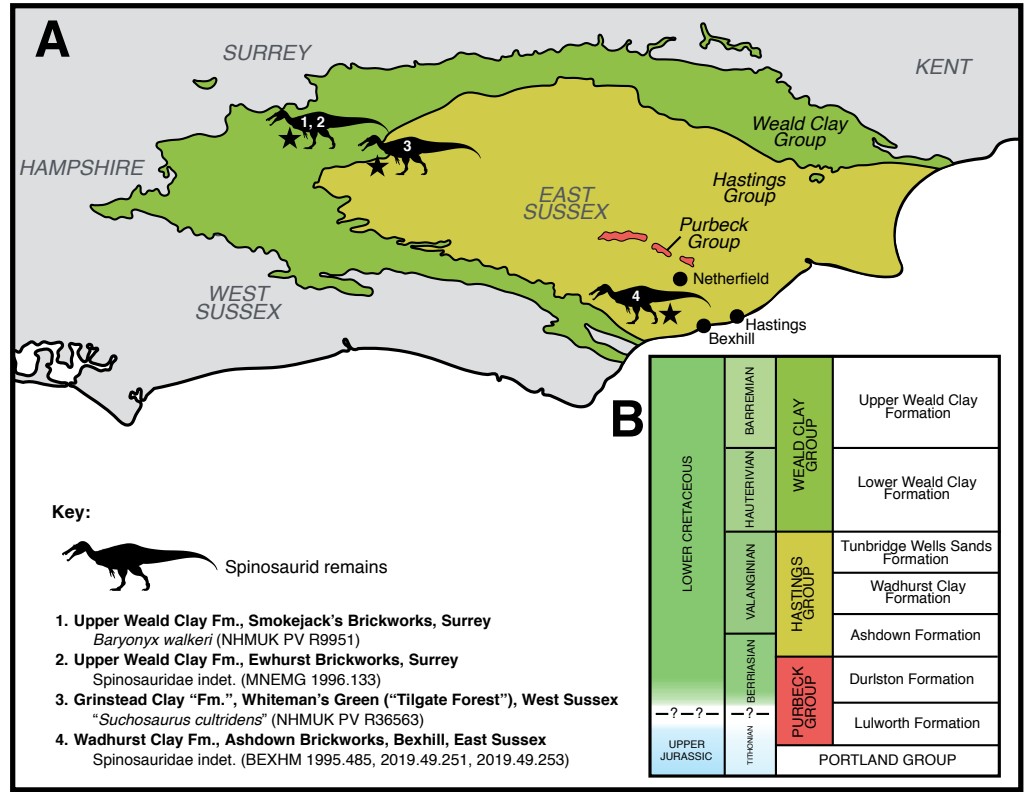

**Figure 1** **Geological context of the Lower Cretaceous deposits of southeast England, focussing on the Purbeck Group and Wealden Supergroup.** (A) Schematic geology of the Lower Cretaceous deposits of the Weald Sub-basin (southeast England), highlighting published spinosaurid finds (*Charig & Milner, 1997*; *Salisbury & Naish, 2011*; *Turmine-Juhel et al., 2019*). Based on *Austen & Batten* (*2018*: Fig. 2). Note that various additional spinosaurid teeth are known from the region but remain undescribed in detail (*Fowler, 2007*). (B) Simplified stratigraphic column of the Weald Group in southeast England, based on *Batten & Austen* (*2011*: Fig. 3.2). Note that the Grinstead Clay Formation, which subdivides the Tunbridge Wells Sands Formation in *Batten & Austen (2011)* and from which the "*Suchosaurus cultridens*" type specimen was discovered (*Salisbury & Naish, 2011*), is downgraded to a member of the latter formation in other works (*Hopson, Wilkinson & Woods, 2008*) and has not been included in this column. Spinosaurid silhouette courtesy of Dan Folkes (CC-BY 4.0).

and spans the Jurassic-Cretaceous boundary (Tithonian–Berriasian; see below). A Purbeck origin for HASMG G369a would be important, as theropods are the rarest terrestrial vertebrate fossils from the Purbeck Group (*Barrett, Benson & Upchurch, 2010*; *Benson & Barrett, 2009*; *Milner, 2002*) and Purbeck spinosaurid remains have not previously been reported.

Isolated theropod teeth are common in the Mesozoic fossil record (*Hendrickx et al., 2019*; *Smith, Vann & Dodson, 2005*) but their identification to lower taxonomic levels has been fraught with issues, among which are rampant homoplasy and a scarcity of sufficiently detailed anatomical accounts (*Hendrickx, Mateus & Araújo, 2015a*; *Hendrickx, Tschopp & Ezcurra, 2020*). Obviously, theropods possess a wide variety of dental morphologies (*Hendrickx & Mateus, 2014*; *Hendrickx, Mateus & Araújo, 2015b*; *Hendrickx et al., 2019*),

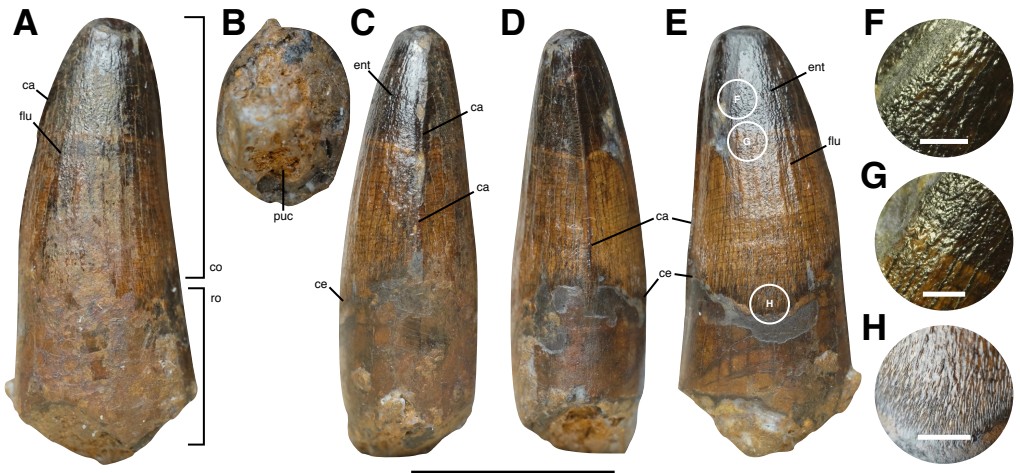

**Figure 2** Isolated tooth HASMG G369a. (A) Lingual, (B) basal, (C) mesial, (D) distal and (E) labial view. (F–G) Close up of the enamel texture on the labial tooth surface. Abbreviations: ca, carina; ce, cervix; co, crown; ent, enamel texture; flu, flute; puc, pulp cavity (infilled); ro, root. Scale bars (A–E): 10 mm, (F–G): 1 mm.

and various characters have the potential to allow the identification of isolated specimens to their respective clades (*Hendrickx, Tschopp & Ezcurra, 2020*). Recent works advocate for the combined use of cladistic, discriminant and cluster methods in order to provide robust support and minimise the misleading impact of homoplasy (*Hendrickx & Mateus, 2014*; *Hendrickx, Tschopp & Ezcurra, 2020*). Here, we aim to identify HASMG G369a *via* the application of these methods, and to test the aforementioned assumption that British spinosaurid material should be considered referable to *Baryonyx* by default. The specimen's provenance is also discussed, and the fossil record of early and post-Cenomanian spinosaurids is reviewed.

## Geological context and provenance of HASMG G369a

The collection of teeth labelled as HASMG G369 consists of 10 specimens, and is associated with a note, which states:

*"If no specific locality is mentioned, these specimens are from Netherfield (Purbeck)"*

No specific locality is mentioned for any of the specimens, and it is unclear when or by whom this note was written. Importantly, the note is inconsistent with the accession record for HASMG G369, which details a "*collection of local Wealden fossils*" gifted by the Reverend Pierre Tielhard de Chardin (1881–1955); the provenance and contents of this "collection" are unknown. Tielhard is known to have collected from the Ashdown and Wadhurst Clay formations around Hastings, and donated many specimens (including some vertebrate remains) to Hastings Museum (*Brooks, 2008*). Thus, within the Weald sub-basin, HASMG G369a was either found from the Purbeck Group near Netherfield or the overlying Wealden Supergroup strata surrounding Hastings (Fig. 1A).

Three fault-bounded inliers result in surface exposures of the Purbeck Group within the Weald sub-basin, located north and northwest of Battle in East Sussex, and are

surrounded by the overlying Hastings Group (most of which comprise deposits of the Ashdown Formation) (*Howitt, 1964*; *Milner, 1922*; *Radley & Allen, 2012a*). These are the oldest exposed rocks in the region, with the inliers located north of Brightling, between Hollingrove and Netherfield, and near Archer Wood (*Lake & Shepard-Thorn, 1987*); the foremost pair have been respectively referred to as the Rounden Wood/Brightling-Heathfield and Limekiln Wood/Mountfield inliers (*Howitt, 1964*; *White, 1928*). The Purbeck Group in the area was previously quarried and mined, with data also provided from boreholes; however, surface exposures are poor and are mainly visible following valley denudation; those exposed in stream valleys have often been disturbed by valley-bulging, landslips and slope cambers (*Lake & Shepard-Thorn, 1987*; *Topley, 1875*). Nevertheless, exposures of the Purbeck Group in the region are represented by both of its constituent Lulworth and Durlston Formations (Fig. 1B), which are principally Berriasian in age (*Cope, 2007*; *Hopson, Wilkinson & Woods, 2008*; *Howitt, 1964*; *Lake & Shepard-Thorn, 1987*). As mentioned above, Purbeck theropods are very rare, and documented specimens from Sussex outcrops include material referred to "*Megalosaurus* sp." (*Benton & Spencer, 1995*; *Topley, 1875*; *White, 1928*).

The Hastings Group, itself the basal unit of the Wealden Supergroup within the Weald sub-basin (*Batten, 2011*), dominates the area surrounding Hasting and is comprised of the older (late Berriasian–early Valanginian) Ashdown Formation, followed by the Wadhurst Clay Formation (Valanginian) and Tunbridge Wells Formation (late Valanginian; Fig. 1B), several of which are well exposed along coastal sections (*Hopson, Wilkinson & Woods, 2008*; *Lake & Shepard-Thorn, 1987*; *Radley & Allen, 2012a*). Only a small outcrop of the overlying Weald Clay Group is known near Cooden (*Lake & Shepard-Thorn, 1987*). Vertebrate fossils from the coastal exposures around Hastings in particular have been collected for over a century (*Benton & Spencer, 1995*). Documented theropod finds from the Hastings area include an allosauroid tibia (HASMG G378) (*Naish, 2003*) and material referred to "*Megalosaurus dunkeri*" (*e.g.*, NHMUK PV R19154) and "*M. oweni*" (*Benton & Spencer, 1995*; *White, 1928*). Allosauroid and spinosaurid teeth are also known from the Wadhurst Clay around Bexhill (*Charig & Milner, 1997*; *Turmine-Juhel et al., 2019*), as are the remains of a tiny maniraptoran (*Naish & Sweetman, 2011*). The enigmatic theropod *Altispinax* (NHMUK PV R1828) is also known from the Hastings Group of Battle (*Maisch, 2016*; *Naish, 2011*; *Von Huene, 1923*), located between Netherfield and Hastings.

We were unable to clarify the conflicting accession information surrounding HASMG G369a or ascertain its provenance. Given the rarity of Purbeck Group theropods, limited exposure of the succession around Netherfield, and accession history, we consider it highly unlikely this tooth originates from the Purbeck Group. Further, in the overlying Hastings Group, vertebrate fossils (bar fish detritus) are also extremely rare in the Ashdown Formation around Hastings and the exposures of the Weald Clay Formation are highly limited (*Lake & Shepard-Thorn, 1987*). Taken together, the upper units of the Hastings Group succession are thus the more likely candidates regarding HASMG G369a's provenance, and we thus provisionally consider the specimen to be Valanginian in age.

## MATERIALS & METHODS

### Orientation and terminology

Dental nomenclature and protocols for crown and denticle morphometry follow the recommendations of *Hendrickx, Mateus & Araújo (2015b)* and references therein.

### Measurements

The specimen was examined *via* a DinoLite (AM4113TL) digital microscope. Measurements were taken using a 150 mm digital calliper (accuracy 0.01 mm), as well as the measurement tools in DinoXcope (v2.0.4) software. A full list of measurements is provided in the Supplementary Information.

As HASMG G369a is missing its apex, several ordinary least-squares regression analyses were conducted where the specimen's crown height (CH) was compared against crown base length (CBL) and crown base width (CBW) for other spinosaurid teeth. Measurements were collected from the dataset of *Hendrickx, Tschopp & Ezcurra (2020)*. Variables were log-transformed to fit a normal distribution and the analyses were conducted using the *Bivariate regression* function (*Model > Linear*) in Past4 (v.4.11) (*Hammer, Harper & Ryan, 2001*). Of the different spinosaurid samples analysed (see Supplementary Information), logCBW from *Baryonyx walkeri* lateral teeth provided the most favourable regression coefficient ( $r^2 = 0.86$ ), the slope and intercept of which was then used to estimate crown height in HASMG G369a. Other measurements or descriptions derived from CH (*e.g.*, mid-crown length and width, number of denticles at mid-crown *etc.*) were based on the estimation detailed above.

Crown angle (CA) was estimated using the *Angle* tool in FIJI (*Schindelin et al., 2012*) *via* the creation of a vertex delimited by the CBL and a line trending through the midpoint of the preserved apex as the specimen was observed in lateral view. The landmarks used to delineate CBL follows *Hendrickx, Mateus & Araújo (2015b)*. *Hendrickx, Mateus & Araújo (2015b)* described a method to calculate CA using the law of cosines and several morphometric landmarks, but photographs and FIJI has also been employed for isolated theropod crowns (*Hendrickx, Tschopp & Ezcurra, 2020*).

### Cladistic analysis

We examined the phylogenetic affinities of HASMG G369a by including it in an updated version of the *Hendrickx & Mateus (2014)* data matrix designed to test the affinities of non-avian theropod teeth (*Hendrickx, Tschopp & Ezcurra, 2020*). This updated matrix was used to assess the affinities of an isolated theropod tooth associated with the *Aerosteon riocoloradensis* holotype: the latter operational taxonomic unit (OTU) was replaced by HASMG G369a, and the final matrix was composed of 146 characters (Ch.) scored across 106 theropod OTUs (the "whole dentition" dataset). The mesial and lateral dentitions of spinosaurids are difficult to distinguish (*Hendrickx, Mateus & Araújo, 2015b*). However, as early spinosaurids possessed supernumerary lateral teeth (*e.g.*, *Baryonyx* NHMUK PV R9951), it is more likely that HASMG G369a originated from the more distal maxillary or dentary dentition. HASMG G369a was thus scored as a lateral tooth.

We performed the cladistic analysis in TNT 1.5 (*Goloboff & Catalano, 2016*) following the methods outlined in *Young et al. (2019)* and *Hendrickx, Tschopp & Ezcurra (2020)*, based on a backbone tree topology and the positive constraint command (*force +*), setting HASMG G369a as a floating terminal. The references used to create the backbone tree can be found in *Hendrickx, Tschopp & Ezcurra (2020)*. A pair of additional cladistic analyses was also performed using the whole dentition dataset without constraints, and a reduced matrix consisting only of crown-based characters (see *Young et al., 2019*; *Hendrickx, Tschopp & Ezcurra, 2020*: 11). The latter included 91 characters (Ch. 38–122 and 141–146) scored for 101 OTUs, with all edentulous taxa removed.

The tree searching strategy involved a combination of algorithms: Wagner trees, TBR branch swapping, sectorial searches, Ratchet (perturbation phase stopped after 20 substitutions) and Tree Fusing (five rounds) were used until 100 hits of the same minimum tree length were reached. The recovered trees were subsequently subjected to an additional round of TBR branch swapping. In the unconstrained analyses, wildcard OTUs were identified using the iterPCR function (*Goloboff & Szumik, 2015*; *Pol & Escapa, 2009*), and Bremer support values were calculated as a measure of nodal support in the resulting reduced consensus.

*Hendrickx & Mateus (2014)* use hypodigms for their spinosaurid OTUs, given the type specimens for several do not preserve dental elements (*e.g.*, *Suchomimus*) or have been lost entirely (*e.g.*, *Spinosaurus*). We note that their *Baryonyx* OTU includes the *B. walkeri* holotype NHMUK PV R9951 and the Iberian specimen ML 1190, and that the latter was recently considered the type specimen of a distinct taxon, *Iberospinus natarioi* (*Mateus & Estraviz-López, 2022*). *Mateus & Estraviz-López (2022)* combined the dental character matrix of *Hendrickx et al. (2020)*—itself a version of the matrix used in the present work—with the modified pan-skeletal matrix of *Arden et al. (2019)* in their phylogenetic analysis of ML 1190. The latter specimen was coded for 36 observable dental characters, however it would appear that *Mateus & Estraviz-López (2022)* did not realise that the *Baryonyx* OTU employed in their analysis is a hypodigm and already contained ML 1190 (*Hendrickx & Mateus, 2014*). Nevertheless, the spinosaurid OTUs used in our analysis of the *Hendrickx, Tschopp & Ezcurra (2020)* matrix were not modified given the fact that the dental material of *I. natarioi* is limited, positionally overlaps with that of *B. walkeri*, and possesses the same (observable) character scores as the *Baryonyx* OTU.

Elsewhere, the OTU of *Irritator* also includes the type specimen of *Angaturama*, following previous authors who consider the latter congeneric with the former (*Buffetaut & Ouaja, 2002*; *Charig & Milner, 1997*; *Dal Sasso et al., 2005*; *Sereno et al., 1998*; *Sues et al., 2002*). Specimens used for the cf. *Suchomimus* and cf. *Spinosaurus* hypodigm OTUs can be found in *Hendrickx & Mateus* (*2014*: Table 1).

Regarding character scores, those of Ch. 82 (concerning the basalmost position of the mesial serration in lateral teeth) were scored by a process of elimination: although the basalmost mesial serration is not preserved in HASMG G369a, it likely possessed state 1 given the preserved extent of the mesial denticles and the probable inapplicability of states 0 and 2. Meanwhile, Ch. 90 (denticle number in lateral teeth respectively) were extrapolated
**Table 1  Measurements of the reconstructed HASMG G369a used in the morphometric analyses.**

| | |
|---|---|
| Crown base length (CBL) | 8.16 |
| Crown base width (CBW) | 7.03 |
| Crown height (CH)[*] | 17.2 |
| Apical length (AL) | ? |
| Midcrown length (MCL)[*] | 5.67 |
| Midcrown width (MCW)[*] | 4.54 |
| Mesial serrated carina length (MSL) | ? |
| Number of labial flutes (+1) (LAF) | 7 (8) |
| Number of lingual flutes (+1) (LIF) | 5 (6) |
| Crown angle (CA) | 74 |
| Mesial denticle length (MDL) | ? |
| Distal denticle length (DDL) | 0.171 |

Notes.

Measurements in millimetres (mm) and crown angle in degrees (°). Asterisk (*) marks measurements derived from reconstructed, rather than observed, crown height (see main text).

from the observable data due to the incomplete nature of the carinae and preservation of denticles.

## Discriminant function analyses
### Pan-theropodan datasets

To classify and predict its optimal classifications inside "family-level" groupings based on quantitative data, HASMG G369a was included in a large published dataset of theropod teeth (*Hendrickx, Tschopp & Ezcurra, 2020*) and subjected to a discriminant function analysis (DFA) in Past4, where it was treated as an unknown taxon and classified at genus or clade levels. Pertinent to this work, the British spinosaurids previously included in this dataset were the type specimens of *Baryonyx walkeri* (NHMUK PV R9951) and "*Suchosaurus cultridens*" (NHMUK PV R36536). As above, HASMG G369a replaced the tooth associated with the *Aerosteon* holotype examined in *Hendrickx, Tschopp & Ezcurra (2020)*. The discriminant function analysis was performed following the protocol detailed by *Young et al. (2019)* and implemented in *Hendrickx, Tschopp & Ezcurra (2020)*, where all variables were log-transformed to normalize the quantitative variables, and a $\log(x + 1)$ correction was applied to LAF and LIF to account for the absence of flutes on the crown, and an arbitrary value of 100 denticles per five mm was used for unserrated carinae (see *Young et al. (2019)* regarding justification of the latter modification).

The final dataset included 1335 teeth belonging to 89 taxa (84 species and five indeterminate family-based taxa) separated into 20 monophyletic or paraphyletic group measured for 12 variables (CBL, CBW, CH, AL, MCL, MCW, MSL, LAF, LIF, CA, MDL, DDL; see Table 1). As noted in *Hendrickx et al. (2020)*, *Young et al. (2019)* and *Hendrickx, Tschopp & Ezcurra (2020)* incorrectly use the abbreviation DCL and DDC for DDL. Due to inconsistencies between authors when measuring dinosaur tooth crowns (*Hendrickx, Tschopp & Ezcurra, 2020*), a second analysis was conducted on a reduced dataset restricted to measurements previously taken by a single author using a consistent measuring protocol.
This reduced dataset includes 594 teeth belonging to 72 theropod taxa separated into 20 monophyletic or paraphyletic groups.

In sum, clade- and genus-level discriminant function analyses were conducted on both the whole and reduced pan-theropodan datasets. These datasets were subject to an additional round of clade- and genus-level analyses where the absence of denticles was considered inapplicable (no denticles = "?").

### Spinosaurid-only datasets

In order to assess the morphospace occupied by each spinosaurid specimen, additional discriminant function analyses were conducted on the raw morphometric data from *Hendrickx, Tschopp & Ezcurra (2020)* focussing only on Spinosauridae. HASMG G369a was thus added to a dataset that included teeth from *Baryonyx*, cf. *Suchomimus*, *Irritator*, "*Sinopliosaurus fusuiensis*" and "*Suchosaurus cultridens*", as well as teeth referred to cf. Baryonychinae (XMDFEC V10010) and various indeterminate Spinosaurinae (the specimens and their associated data are compiled from *Hendrickx, Tschopp & Ezcurra (2020)*; see Supplementary Information). Only teeth from *Baryonyx*, *Irritator*, "*Suchosaurus*" and "*Sinopliosaurus*" are from holotype specimens.

We follow *Hendrickx, Mateus & Araújo (2015a)* in performing two analyses in Past4 where all morphometric variables of interest ($n = 35$) were included in the first instance, followed by an analysis were ratio variables (MAVG, DAVG, CBR, CHR, MCR, MEC, DSDI, CA, CDA, CMA and CAA) were excluded; CDA is derived from two ratio variables (*Richter, Mudroch & Buckley, 2013*) and thus also excluded from this second analysis. The variables "transverse undulations" and "interdenticular sulci" were excluded from both analyses as the former contained qualitatively described data whilst the presence of the latter is not a character associated with spinosaurid dentition (*Hendrickx et al., 2019*). Alternative versions of variables (*i.e.,* CA2, DAVG2), were also excluded so as not to inflate the dataset.

As in *Hendrickx, Mateus & Araújo (2015a)*, measurements were not log-transformed. Missing or uncertain data were coded as "?", whilst characters with an uncertain data range were averaged (*e.g.,* the value 11.5 was used for the "11 or 12" lingual flutes scored for "*Suchosaurus*" NHMUK PV R36536). Data prefaced with a greater or less than sign were arbitrarily adjusted by plus or minus one point respectively (*i.e.,* ">5" was changed to "6"). Data scored as "absent" or "not applicable" (represented by a dash) were replaced with the value zero. The "absent?" data point for the lingual flutes of cf. *Suchomimus* specimen UC G73-3 was changed to "?" given the uncertainty of the interpretation. These changes are compiled with the Supplementary Information.

A second round of analyses was undertaken, based on a reduced spinosaurid sample excluding the *nomina dubia* "*Suchosaurus*" (NHMUK PVR 36536) and "*Sinopliosaurus fusuiensis*" (IVPP V4793.1), as well as cf. Baryonychinae (XMDFEC V10010) given suggestions this specimen does not represent a spinosaurid taxon (see also below) (*Buffetaut et al., 2019*; *Katsuhiro & Yoshikazu, 2017*; *Soto, Toriño & Perea, 2020*). The remaining spinosaurids were subjected to the same analyses described above (*i.e.,* one DFA using all variables and another excluding ratio variables).

## Cluster analysis

Cluster analyses were also performed in Past4 on the different pan-theropodan datasets mentioned above. Hierarchical clustering with a Paired group algorithm and Neighbour joining clustering were used, rooting the tree with the final branch, whilst selecting Euclidean distances as the similarity index.

# RESULTS

## Systematic palaeontology

DINOSAURIA *Owen 1842*
THEROPODA *Marsh 1881*
TETANURAE *Gauthier 1986*
SPINOSAURIDAE *Stromer 1915*
Spinosauridae gen. and sp. indet.

## Description

### Orientation

The slight distal recurvature of the crown means that HASMG G369a can be oriented along its mesiodistal axis but the labiolingual axis is less clear. A basal depression, ordinarily lingually situated in theropods (*Hendrickx, Mateus & Araújo, 2015b*), is absent on either side of HASMG G369a. This crown subunit may appear planar in some theropods (*Hendrickx, Mateus & Araújo, 2015b*), but this is also not the case in HASMG G396a. The crown does, however, display slight labiolingual curvature when viewed distally, and we use this feature to differentiate the lingual and labial surfaces.

### Condition

HASMG G369a comprises a near-complete crown (lacking its apex) associated with the basal portion of the root. The enamel is largely well preserved on the labial surface excepting a small chip apically. Large parts of the enamel on the lingual surface however have been worn.

The preserved mesial carina has been abraded in several places, such that only two short sections remain: one just above the cervix and the other located mesiocentrally; the denticles—where preserved—appear slightly worn. The distal carina is more complete, with wear mainly affecting the apical-most portion.

### Crown

HASMG G369a is a conidont crown with a lenticular cross section at the cervix and at mid-crown (Figs. 2A–2B); as such, the crown is weakly labiolingually compressed (CBR: 0.86). The crown is not particularly large (preserved CH: 13.2 mm; reconstructed CH: 17.2 mm) and only moderately elongated (preserved CHR: 1.68; reconstructed CHR: 2.1).

The mesial and distal carinae are both denticulated (Figs. 2C–2D, 3), lacking adjacent concave surfaces. The former is straight, undivided, and not notably developed, and is positioned largely centrally on the mesial profile. Whilst the basalmost portion has been chipped off (see above), what remains suggests the mesial carina almost certainly reached

the cervix. The distal carina is slightly diagonally oriented, and as mentioned above, trending towards the labial side basally. It too is not markedly developed and lacks any twisting of splitting. It extends basally past the cervix a short distance. The apical extent of either carina cannot be determined for this specimen.

The crown displays weak distal recurvature in labiolingual views (Fig. 2A). Its mesial profile is weakly convex, whilst the distal profile is almost straight for the majority of its preserved length. The preserved apex is almost centrally positioned. When viewed distally (Fig. 2D), the crown also possesses minor lingual curvature, with the apex closer to the lingual side. Both the labial and lingual crown surfaces are convex.

The cervix assumes a parabolic morphology on the better-preserved labial side of the crown, such that the basalmost extent of the enamel occurs roughly centrally (Fig. 2E). The equivalent features, or relative extent of the enamel on the lingual side, cannot be reliably ascertained due to preservation. However, the extent of the enamel on the mesial and distal surfaces appears largely similar.

### Denticles

The denticles (Fig. 3) of the mesial carina are best preserved at mid-crown, although some incipiently visible ones are also observed at the basalmost preserved portion of the carina. Those of the distal carina are present across a large extent but are worn distoapically and between the distocentral and distobasal portion of the crown.

There are approximately seven denticles per millimetre on both the mesial and distal carinae at midcrown. These are typically mesiodistally longer than apicobasally tall and are oriented perpendicularly relative to their respective carina. Their external margins are flattened, giving them a mesiodistally subrectangular appearance in lateral view. The interdenticular spaces are relatively broad and well developed, though the interdenticular diaphyses are not easily recognised, perhaps due to preservation. The mesial and distal denticles at midcrown are approximately the same size (denticle size density index (DSDI): 1), and interdenticular sulci are not observed on either carina. The more complete distal carina also reveals a regular variation in denticle size; this attribute can also be extended to those sections preserved on the mesial carina.

The basalmost segments of the carinae are also denticulated. However, those present mesially are difficult to measure and describe, being visible only under certain light conditions and orientations. Those situated distobasally appear to extend to the cervix (if not just beyond the latter) and are generally similar to those of the midcrown, being smaller and slightly more numerous per millimetre.

### Ornamentations

The crown is ornamented, possessing weakly developed flutes, of which seven (possibly eight) are present on the lingual side and five on the labial one (Figs. 2A and 2E). Those adorning the latter surface are less prominent. Transverse and marginal undulations appear absent. The crown possesses veined enamel texture basally, which is particularly fine near the cervix and whose grooves/ridges are generally apicobasally oriented barring those that curve towards the carina. More apically however, the texture becomes irregular (Figs. 2F–2H).

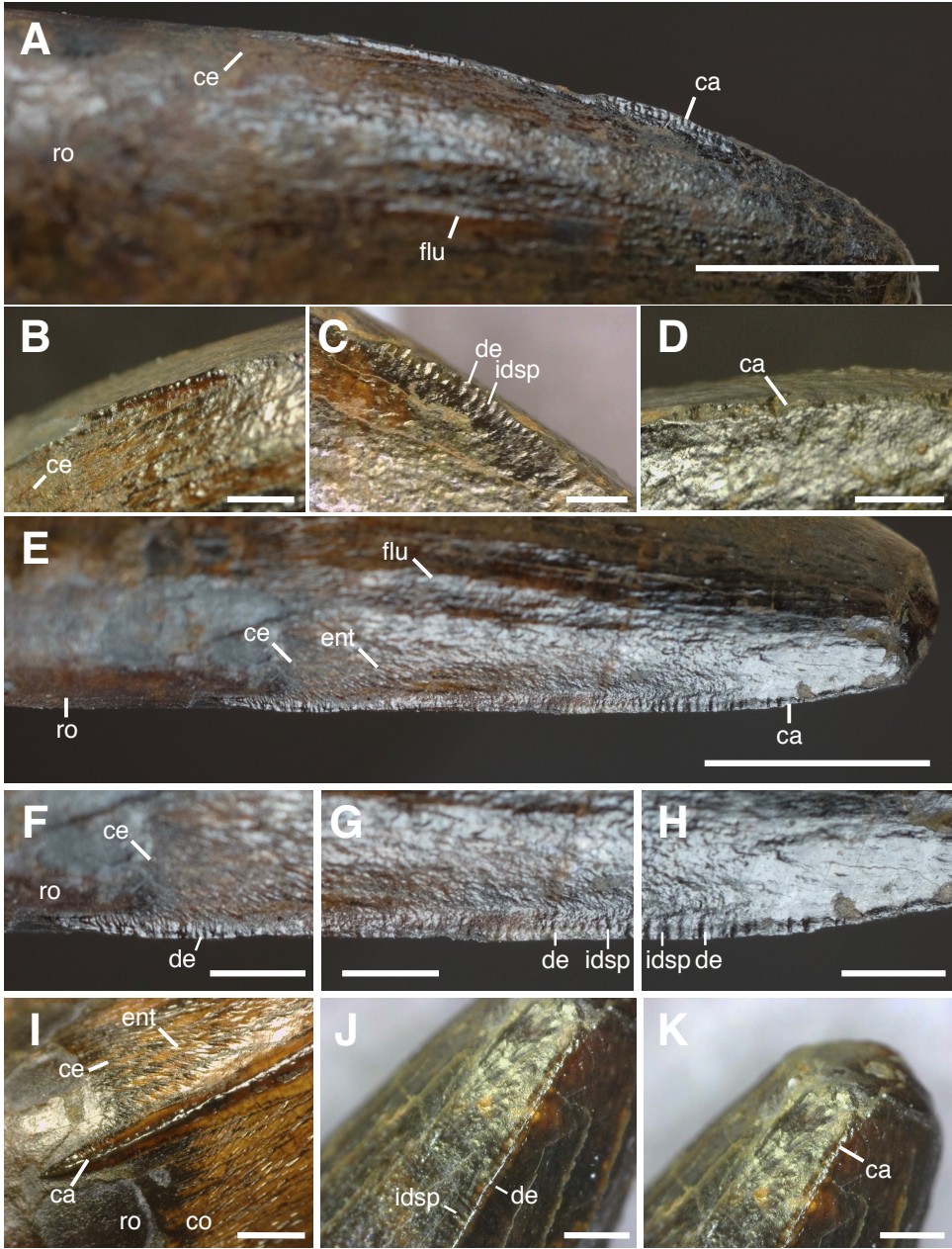

**Figure 3** **Close up of carinae and denticles of HASMG G369a.** Mesial carina in (A) lateral view. Close up of mesial carina in (B) mesiobasal, (C) mesiocentral and (D) mesioapical views. Distal carina in (E–H) lateral and (I–J) distal views. Close up of (F) distobasal carina, (G) distocentral carina, (H) distoapical carina. Abbreviations: ca, carina; ce, cervix; co, crown; de, denticle; ent, enamel texture; flu, flute; idsp, interdenticular space; ro, root. Scale bars: (A, E) 5 mm, (B–D, F–K) 1 mm.

### Cladistic analysis

The results of the various cladistic analyses, detailed below, are summarised in Table 2. Full versions of the recovered trees are available in the Supplementary Information.

### Whole dentition dataset

Two MPTs of 1318 steps were recovered following the constrained search on the whole dentition dataset (CI = 0.204097, RI = 0.451360). HASMG G369a either assumed a position outside the baryonychine + spinosaurine clade or at the base of Spinosaurinae; the latter position was supported by a single synapomorphy: a slightly convex mesial margin (Ch. 73:1). Accordingly, the strict consensus recovered HASMG G369a in a polytomous Spinosauridae alongside Baryonychinae and Spinosaurinae (Fig. 4A), with the clade supported by numerous synapomorphies. Of these, HASMG G69a shared: (1) weak labiolingual compression of the crown with a CBR exceeding 0.75 (Ch. 70:2), (2) subcircular basal cross-section of the crown (Ch. 76:0), (3) over 30 distocentral denticles per five mm (Ch. 89:0), (4) fluted enamel surfaces present on both labiolingual surfaces (Ch. 111:2) and (5) veined enamel texture (Ch. 121:3).

The unconstrained search on the whole dentition dataset initially returned 248 MPTs of 1074 steps (CI = 0.250466, RI = 0.578975). This increased to 87576 MPTs following the round of TBR. The strict consensus is largely unresolved and predominantly formed by two large polytomies containing well over 25 OTUs each. Few traditional clades can be recognised but those present include Spinosauridae and Abelisauridae. The strict consensus nevertheless recovered HASMG G369a within a polytomous Spinosauridae alongside Baryonychinae and Spinosaurinae.

A reduced consensus was achieved following the pruning of 23 wildcard OTUs (*Limusaurus* (juvenile), *Masiakasaurus*, *Indosuchus*, *Chilesaurus*, *Piatnitzkysaurus*, *Sciuruminus*, *Eustreptospondylus*, *Afrovenator*, *Dubreuillosaurus*, *Duriavenator*, *Sinraptor*, *Allosaurus*, *Orkoraptor*, *Acrocanthosaurus*, *Aorun*, *Guanlong*, *Eotyrannus*, *Raptorex*, *Gorgosaurus*, *Alioramus*, *Daspletosaurus*, *Tyrannosaurus* and *Ornitholestes*) identified *via* the iterPCR function (Fig. 4B). As above, HASMG G369a is again recovered in a polytomous Spinosauridae alongside Spinosaurinae and Baryonychinae, which is supported by several synapomorphies; those present in HASMG G369a are: (1) the basalmost denticle on the mesial carina of lateral teeth extending to the base of the crown or slightly above the cervix (Ch. 82; see comment in the "Cladistic analysis" methodology section above), (2) basalmost serration on the distal carina situated below the cervix (Ch. 85), and (3) flutes present on both labial and lingual surfaces (Ch. 111).

### Crown-based dataset

The unconstrained search on the crown-based dataset initially recovered 244 MPTs of 648 steps (CI = 0.251543, RI = 0.62139). The additional round of TBR returned over 99999 trees found (overflow). The strict consensus produced a huge polytomy incorporating the vast majority of OTUs including HASMG G369a (see Supplementary Information for the full result). HASMG G369a was one of 74 OTUs acting as wildcard taxa (the others include: *Daemonosaurus*, *Eodromaeus*, *Eoraptor*, *Dracovenator*, *Coelophysis*,

**Table 2    Summary of the cladistic analyses, describing the position of HASMG G369a in Newick format.**

| Dataset | Tooth position | Constrained | Unconstrained | |
| --- | --- | --- | --- | --- |
| | | | Strict consensus | Reduced consensus |
| Whole dentition | Lateral | (HASMG G369a, Spinosaurinae, Baryonychinae) | (HASMG G369a, Spinosaurinae, Baryonychinae) | (HASMG G369a, Spinosaurinae, Baryonychinae) |
| Crown only | Lateral | – | Polytomy with majority of theropod OTUs | n/a |

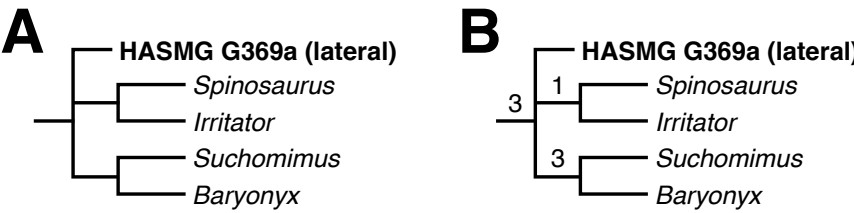

**Figure 4    Results of the phylogenetic analyses.** (A) Strict consensus of the analysis using the whole dataset under constrained conditions. (B) Reduced consensus of the unconstrained analysis using the whole dataset. Numbers at nodes indicate Bremer supports values. Full results can be found in the Supplementary Information.

*Liliensternus, Dilophosaurus, Ceratosaurus, Genyodectes, Berberosaurus, Masiakasaurus, Kryptops, Rugops, Abelisaurus, Aucasaurus, Arcovenator, Chenanisaurus, Indosuchus, Majungasaurus, Skorpiovenator, Piatnitzkysaurus, Marshosaurus, Monolophosaurus, Sciuriminus, Eustreptospondylus, Afrovenator, Dubreuillosaurus, Duriavenator, Megalosaurus, Torvosaurus, Baryonyx, Suchomimus, Irritator, Spinosaurus, Erectopus, Sinraptor, Allosaurus, Neovenator, Fukuiraptor, Australovenator, Megaraptor, Orkoraptor, Acrocanthosaurus, Eocarcharia, Carcharodontosaurus, Giganotosaurus, Mapusaurus, Bicentenaria, Aorun, Zuolong, Proceratosaurus, Guanlong, Dilong, Compsognathus, Ornitholestes, Haplocheirus, Eshanosaurus, Falcarius, Jianchangosaurus, Segnosaurus, Erlikosaurus, Incisivosaurus, Halszkaraptor, Sinornithosaurus, Graciliraptor, Dromaeosaurus, Bambiraptor, Tsaagan, Velociraptor, Sinusonasus, Zanabazar, Troodon* and *Archaeopteryx*).

## Discriminant function analysis
## Pan-theropodan datasets

The analyses conducted on the whole dataset (Fig. 5), regardless of whether the absence of denticles was considered inapplicable or not, consistently classified HASMG G369a as a spinosaurid (clade-level analyses) or referred the tooth to the baryonychine spinosaurid *Suchomimus* (genus-level analyses) (Table 3). Reclassification rates (RR) are, however, generally low, ranging between 59.37–62.07%. Similarly, the reduced datasets based on single-author measurements classified HASMG G369a as a spinosaurid and as *Suchomimus* in the respective analyses (again, with low RR between 59.19–63.74%).

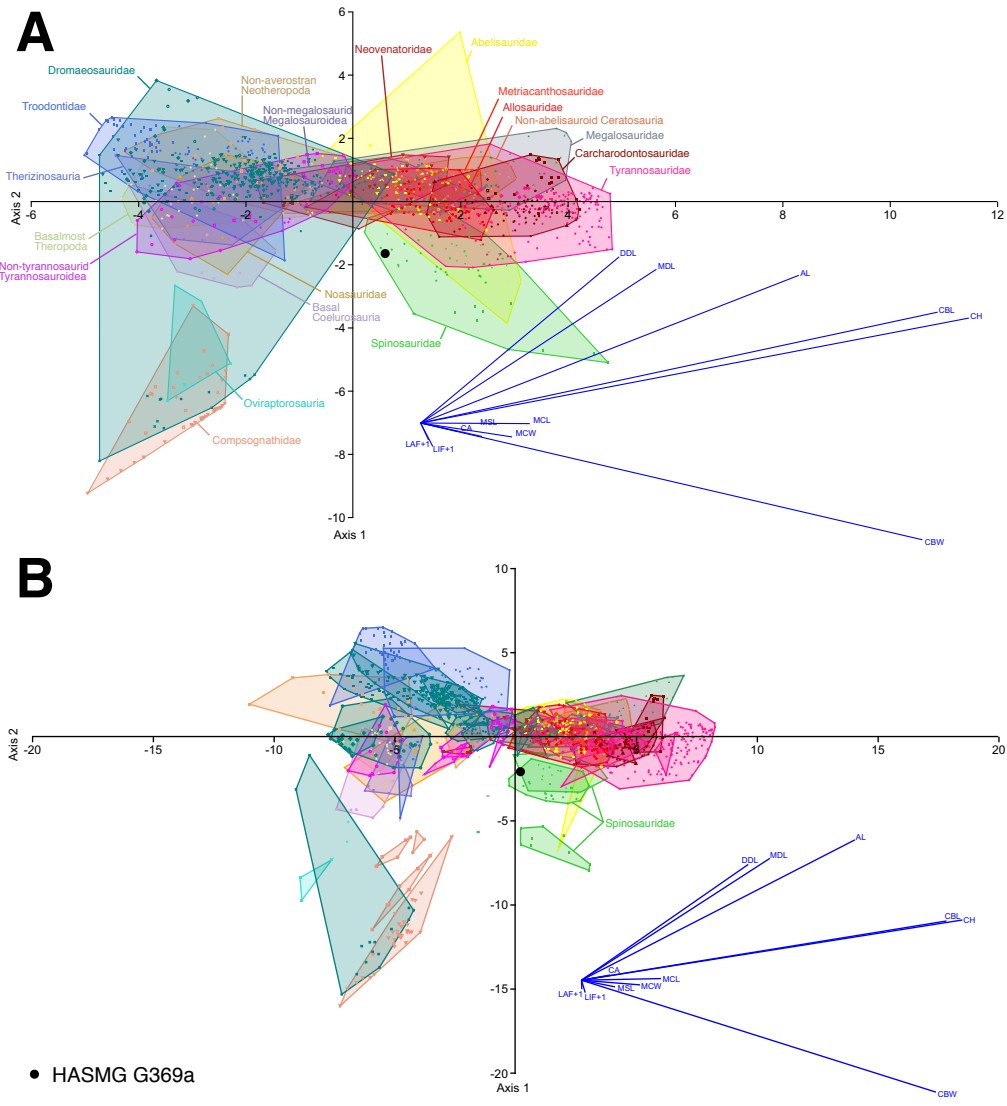

**Figure 5** **Select results of the discriminant function analysis of the pan-theropodan dataset plotted along the first two canonical axes of maximum discrimination in the dataset.** (A) Clade level analysis (eigenvalue of Axis 1 = 5.7073, which accounts for 51.01% of the total variation; eigenvalue of Axis 2 = 2.2155, which accounts for 19.8% of the total variation) and (B) taxon level analysis (eigenvalue of Axis 1 = 18.377, which accounts for 41.04% of the total variation; eigenvalue of Axis 2 = 9.6544, which accounts for 21.56% of the total variation), on the whole dataset consisting of 1335 crowns belonging to 89 taxa (*i.e.,* 84 species and five indeterminate family-based taxa) separated into 20 monophyletic or paraphyletic groups. 61.02% and 61.17% of the theropod specimens were correctly classified to their respective groups and taxa, with HASMG G369a (black dot) respectively classified as a spinosaurid and *Suchomimus* at the clade and taxon-level. Abbreviations: AL, apical length; CA, crown angle; CBW, crown base width; CH, crown height; DDC, distal denticle length; LAF + 1, number of labial flutes plus one; LIF + 1, number of lingual flutes plus one; MCL, mid-crown length; MCW, mid-crown width; MDL, mesial denticle length.

### Spinosaurid-only datasets

The DFA results for the spinosaurid-only morphometric datasets (Table 4) consistently classified HASMG G369a as a non-*Baryonyx* spinosaurid. Reclassification rates are very high (98.18–100%), especially in comparison to the pan-theropodan datasets used above, with HASMG G369a classified as cf. *Suchomimus* in the majority of analyses (PC1 63.73–84.32%, PC2 14.84–26.12%). Interestingly, the results from the dataset including all spinosaurids and all variables classified HASMG G369a as "*Suchosaurus*" (PC1 72.53%, PC2 20.03%), which is also known from the Hastings Group.

Visualisation of the DFA plots also shows that spinosaurid teeth are readily differentiable based on the data from *Hendrickx, Tschopp & Ezcurra (2020)* (Figs. 6 and 7): spinosaurine and baryonychine taxa occupy different morphospace areas, whilst *Baryonyx* and cf. *Suchomimus* do not overlap in any iteration of the analyses. This suggests that *Baryonyx* and cf. *Suchomimus* teeth are morphologically distinct. Whether this impacts discussions regarding the congeneric status of the two taxa remains to be seen, especially given the non-cranial nature of the *Suchomimus* holotype skeleton (*Carrano, Benson & Sampson, 2012*; *Sereno et al., 1998*). Also of note is the tendency for "*Suchosaurus*" to cluster closely with the cf. *Suchomimus* morphospace in the analyses containing all spinosaurid specimens, whilst "*Sinopliosaurus*" plotted close to the morphospace occupied by spinosaurine teeth.

As an aside, the isolated specimen XMDFEC V10010 from the Santonian (Late Cretaceous) Majiacun Formation of China, referred to Baryonychinae by *Hone, Xu & Wang (2010)*, does not cluster closely or share morphospace with any spinosaurid taxon in the DFA analyses of the spinosaurid sample. To explore this further, we tested the specimen using discriminant function and cluster analyses on the "whole", "personal" and "large crown" pan-theropodan datasets from *Hendrickx, Tschopp & Ezcurra (2020)*, treating XMDFEC V10010 as an unknown taxon. These results are presented in full in the Supplementary Information and are briefly discussed below.

### Cluster analysis

The cluster analyses based on the pan-theropodan dataset (Table 5, Supplementary Information), regardless of the method employed (*i.e.,* hierarchical *vs.* neighbour joining), unanimously support spinosaurid affinities of HASMG G369a. Almost all results recover the crown as a sister taxon to *Suchomimus*, except for the Neighbour joining analysis performed on the whole dataset (no denticles = "?"), where it is recovered as sister to a clade containing *Irritator* + *Suchomimus*.

## DISCUSSION

### Affinities of HASMG G369a and the diversity of British spinosaurids

The results from the cladistic, discriminant and cluster analyses clearly support the spinosaurid affinities of HASMG G369a. HASMG G369a shares multiple dental characters in common with spinosaurids, including a sub-circular outline, fluted enamel ornamentation and veined enamel texture, extension of the mesial carina to the cervix and a centrally positioned distal carina (*Hendrickx et al., 2019*).

Of particular note is the finding that HASMG G369a (its wildcard status within the crown-only phylogenetic analyses excepting) failed to associate with *Baryonyx* in any data run. This further supports previous arguments that the Wealden Supergroup contains multiple spinosaurid lineages (*Barker et al., 2021*; *Buffetaut, 2010*; *Naish, 2011*; *Naish & Martill, 2007*). These results also suggest that the spinosaurid diversity within the Wealden Supergroup reflects the situation of coeval Iberian localities, which appear to have contained a more diverse spinosaurid fauna than previously assumed (*Isasmendi et al., 2020*; *Malafaia et al., 2020*; *Mateus & Estraviz-López, 2022*).

The dentition of *Ceratosuchops* and *Riparovenator* were not scored for this analysis due to poor preservation; however, future work should aim to use cladistic and discriminant methods on spinosaurid crowns found in known strata within the Wealden Supergroup in order to further assess the diversity of its spinosaurids. It would be of particular interest to examine isolated spinosaurid teeth from the Upper Weald Clay Formation, in order to test whether these can be confidently referred to *Baryonyx*. Revisiting coeval Lower Cretaceous localities from Iberia may also be useful given the widespread presence of spinosaurids in these deposits (*Malafaia et al., 2020*); several morphometric-based (PCA and DFA) analyses have already been undertaken on Iberian spinosaurid crowns (the results of which also hint at high spinosaurid diversity) (*Alonso & Canudo, 2016*; *Alonso et al., 2018*; *Isasmendi et al., 2020*). However, cladistic analyses are recommended (if not preferred) for the identification of isolated theropod teeth (*Hendrickx, Tschopp & Ezcurra, 2020*), although some alternative machine learning techniques (*e.g.*, decision trees) may be attractive tools with which to assess morphometric data from isolated theropod teeth (*Wills, Underwood & Barrett, 2021*). It should be noted that performing cladistic analyses on single teeth can be time consuming: each individual tooth in a batch of "unknown" specimens has to be tested separately, or appropriately grouped into morphotypes (*Hendrickx, Tschopp & Ezcurra, 2020*). This is further exacerbated by the difficulty distinguishing the position of isolated spinosaurid teeth (*Hendrickx, Mateus & Araújo, 2015b*); whilst we believe a lateral position for HASMG G369a is a more likely origin (see above), spinosaurid samples could alternatively be tested in both positions. Another potential technique for investigating spinosaurid diversity in the Wealden Supergroup might be to conduct specimen-level phylogenetic analyses using Bayesian methods and incorporating stratigraphical information, a method inspired by *Cau (2017)*.

## Comparative anatomy

The large number of minute denticles recalls the condition present in baryonychine spinosaurids (*Hendrickx et al., 2019*). The presence of minute denticles on both carinae most recalls the situation of other British spinosaurid crowns, including those of *Baryonyx* (*Charig & Milner, 1997*), *Riparovenator* (*Barker et al., 2021*), and BEXHM 1995.485 (C. Barker, pers. obs., 2022; *Charig & Milner (1997)* misreported the accession number of this specimen as "BEXHM 1993.485"); the carinae of *Ceratosuchops* are poorly preserved and its dentition will be revisited elsewhere, but denticles are present on some distal carinae at least. The denticles of the "*Suchosaurus cultridens*" type specimen (NHMUK PV R36536) are difficult to discern but this is probably due to wear (*Buffetaut, 2010*). Nevertheless,

Barker et al. (2023), *PeerJ*, DOI 10.7717/peerj.15453

**Table 3** **Results of the discriminant function analyses on the various iterations of the pan-theropodan dataset, with HASMG G369a treated as an unknown taxon.**

| Dataset | Discriminant function analysis | | Reclassification rate (RR) | | Clade level | | Genus level | | Clade level (Eigenvalue) | | Genus level (Eigenvalue) | |
|---|---|---|---|---|---|---|---|---|---|---|---|---|
| | Clade level | Genus level | Clade level (%) | Taxon level (%) | PC1 (%) | PC2 (%) | PC1 (%) | PC2 (%) | Axis 1 | Axis 2 | Axis 1 | Axis 2 |
| Whole dataset | Spinosauridae | *Suchomimus* | 61.02 | 61.17 | 51.01 | 19.8 | 41.04 | 21.56 | 5.71 | 2.22 | 18.38 | 9.65 |
| Whole dataset (no denticles = ?) | Spinosauridae | *Suchomimus* | 62.07 | 59.37 | 50.2 | 19.04 | 42.87 | 17.08 | 5.79 | 2.20 | 18.01 | 7.18 |
| Reduced dataset | Spinosauridae | *Suchomimus* | 59.36 | 63.74 | 57.1 | 21.9 | 41.07 | 24.72 | 12.19 | 4.67 | 24.99 | 15.04 |
| Reduced dataset (no denticles = ?) | Spinosauridae | *Suchomimus* | 59.19 | 60.37 | 54.27 | 22.94 | 41.4 | 25.66 | 10.98 | 4.64 | 23.75 | 14.72 |

**Table 4 Results of the discriminant function analyses on the various iterations of the spinosaurid-only dataset, with HASMG G369a treated as an unknown taxon.**

| Dataset | Discriminant function analysis | Reclassification rate (RR) (%) | Taxon level | | Taxon level (Eigenvalue) | |
|---|---|---|---|---|---|---|
| | | | PC1 (%) | PC2 (%) | Axis 1 | Axis 2 |
| All spinosaurid dataset | "Suchosaurus" | 98.28 | 72.53 | 20.03 | 89.905 | 24.824 |
| All spinosaurid dataset no ratios | Suchomimus | 98.28 | 63.73 | 26.12 | 40.277 | 16.506 |
| Reduced spinosaurid dataset | Suchomimus | 100 | 84.32 | 14.84 | 73.009 | 12.846 |
| Reduced spinosaurid dataset no ratios | Suchomimus | 98.18 | 82.02 | 17.34 | 36.934 | 7.807 |

HASMG G369a differs from some Iberian spinosaurid teeth where a baryonychine dental morphotype lacking mesial denticles has been reported (*Isasmendi et al., 2020*).

Sporadic variation in denticle size is noted in baryonychines and is particularly developed in *Baryonyx* and *Iberospinus* (*Hendrickx et al., 2019*; *Mateus & Estraviz-López, 2022*). In contrast, those of cf. *Suchomimus* change more gradually and sporadic variation in denticle size is mainly observed on the basal portions of the teeth (*Hendrickx et al., 2019*). Those of the preserved mesial dentition of *Riparovenator* are similarly regular (C. Barker, pers. obs., 2022), as are baryonychine teeth from the Barremian–lower Aptian Cameros Basin of Spain (*Isasmendi et al., 2020*). HASMG G369a mirrors the latter specimens in this regard, with the more complete distal carina possessing a largely gradual change of denticle size.

Although damaged in its basal portion, the mesial carina likely reaches or terminates very near the cervix in HASMG G369a, as is common for spinosaurids generally (*Hendrickx et al., 2019*). However, a few spinosaurid crowns, notably from Lower Cretaceous Iberian deposits, do display shorter carinae that extend over only half or two-thirds of the crown height (*Canudo et al., 2008*; *Hendrickx et al., 2019*; *Isasmendi et al., 2020*). A similar feature is also seen in *Iberospinus* (*Mateus & Estraviz-López, 2022*). *Charig & Milner (1997)* described the carinae of BEXHM 1995.485 as failing to reach the cervix, however it would appear that the carinae have been chipped in places, and what remains basally seems to extend past the cervix.

Fluted enamel is typical of spinosaurid crowns (*Hendrickx et al., 2019*), and some have noted that these tend to be more numerous and better developed on the lingual surface (*Buffetaut, 2012*), further corroborating the orientation of the specimen described above. Those present on HASMG G369a, whilst generally weakly developed, are nevertheless in the range of several other spinosaurids: *Baryonyx* and cf. *Suchomimus* average around 6–7 flutes (range 4–8 and 2–10 respectively), whilst an average of 7–8 flutes are observed in *Irritator* (range 5–10) (*Hendrickx et al., 2019*). A similar range (3–9 flutes) has been observed in spinosaurid crowns from Lower Cretaceous Iberian localities (*Ruiz-Omeñaca et al., 2005*). However, the number of flutes in HASMG G369a differs from "*Suchosaurus*" (10–12 flutes) and several spinosaurines (17–20 flutes) (*Hendrickx et al., 2019*). The presence of flutes on both sides of the tooth also makes HASMG G369a different from *Baryonyx walkeri* (where the flutes are almost entirely lingually located), and is instead similar to the condition present in *Ceratosuchops*, *Riparovenator*, "*Suchosaurus*" and cf. *Suchomimus*.

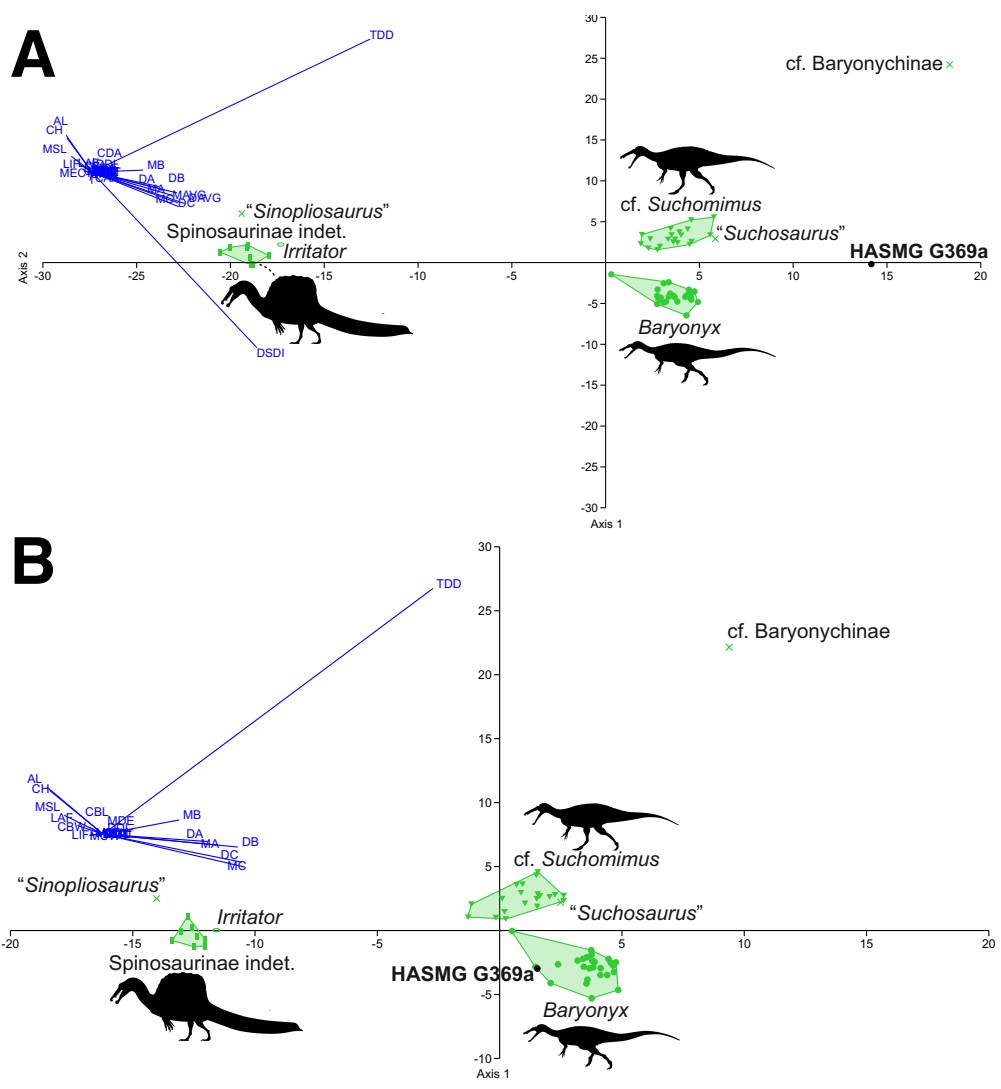

**Figure 6** **Graphical results of the discriminant analyses using a spinosaurid-only dataset comprised of 59 teeth from seven taxa (*Baryonyx*, cf. *Suchomimus*, *Irritator*, Spinosaurinae indet., cf. Baryonychinae, "*Suchosaurus*", "*Sinopliosaurus*").** (A) Results of the analysis including all variables (PC1 72.53, PC2 20.03; eigenvalue of axis 1: 89.905, axis 2: 24.824; reclassification rate = 98.28%), where HASMG G369a was referred to "*Suchosaurus*". (B) Results of the analysis excluding ratio variables (PC1 63.73, PC2 26.12; eigenvalue of axis 40.277, axis 2: 16.506; RR = 98.28%), where HASMG G369a was referred to cf. *Suchomimus*. Abbreviations: see *Hendrickx, Mateus & Araújo (2015b)* and *Richter, Mudroch & Buckley (2013)*. Phylopic silhouette credits: Spinosaurinae indet.: Ivan Iofrida (CC-BY-4.0, https://creativecommons.org/licenses/by/4.0/); *Baryonyx* and *Suchomimus*: Scott Hartman (CC-BY-NC-SA-3.0).

Other forms of enamel ornamentation, such as the transverse undulations observed in some *Baryonyx* (NHMUK PV R9951), *Iberospinus* (ML1190) and cf. *Suchomimus* crowns (*e.g.*, MNN G67-1), or the marginal undulations present in *Baryonyx*, *Irritator* (SMNS 58022), cf. *Suchomimus* (*e.g.*, MNN G35-9) and indeterminate Brazilian spinosaurines (*Hendrickx et al., 2019*; *Hendrickx, Tschopp & Ezcurra, 2020*; *Medeiros, 2006*), are absent

 

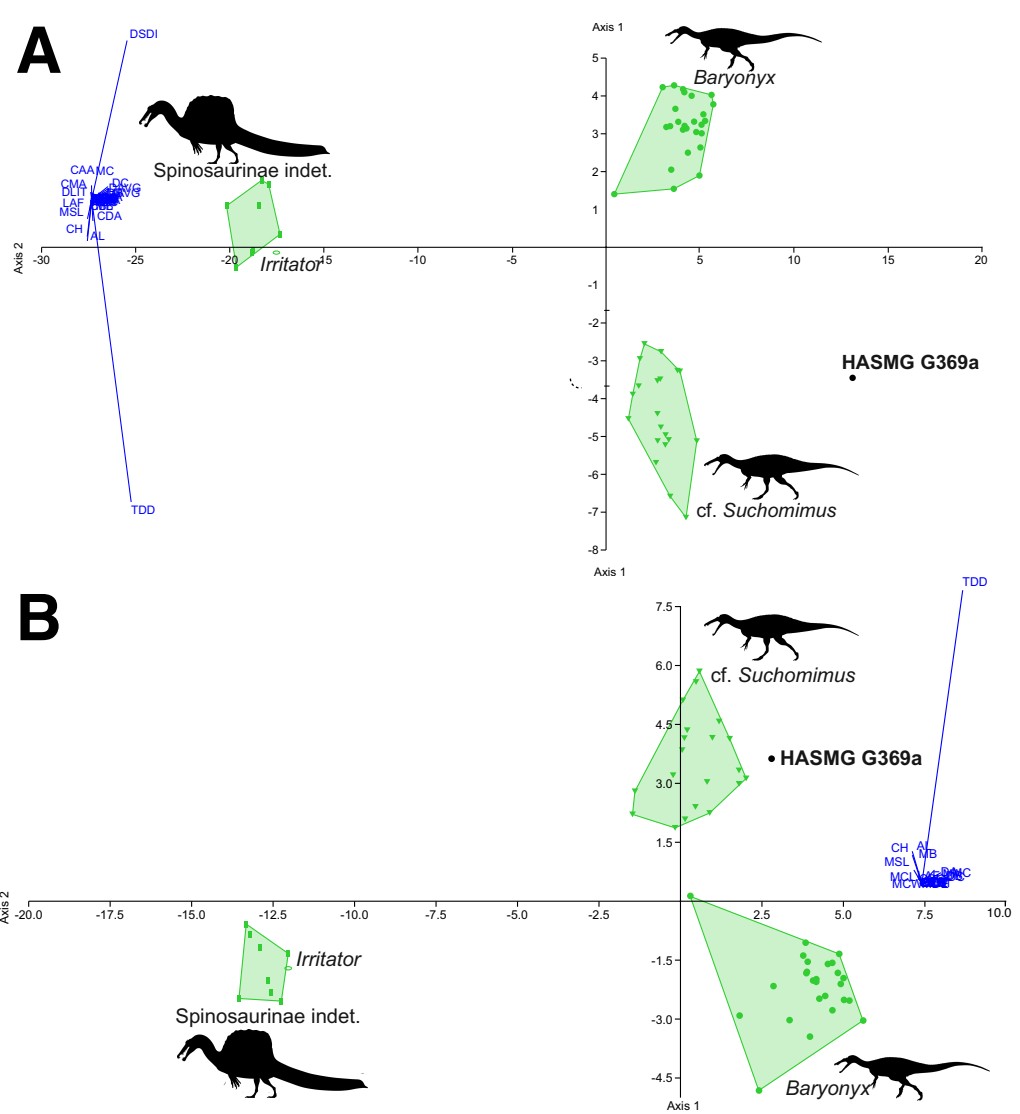

**Figure 7** Graphical results of the discriminant analyses using a spinosaurid-only dataset comprised of 56 teeth from 4 taxa (***Baryonyx***, cf. ***Suchomimus***, ***Irritator***, **Spinosaurinae indet.**), including **HASMG G369a as an unknown taxon.** (A) Results of the analysis including all variables (PC1 84.32, PC2 14.84; Eigenvalue of axis 1: 73.009, axis 2: 12.846; reclassification rate = 100%), where HASMG G369a was referred to cf. *Suchomimus*. (B) Results of the analysis excluding ratio variables (PC1 82.02, PC2 17.34; Eigenvalue of axis 36.934, axis 2: 7.807; RR = 98.18%), where HASMG G369a was referred to cf. *Suchomimus*. Abbreviations: see (*Hendrickx, Mateus & Araújo, 2015b*; *Richter, Mudroch & Buckley, 2013*). Phylopic silhouette credits: Spinosaurinae indet.: Ivan Iofrida (CC-BY-4.0, https://creativecommons.org/licenses/by/4.0/); *Baryonyx* and *Suchomimus:* Scott Hartman (CC-BY-NC-SA-3.0, https://creativecommons.org/licenses/by-nc-sa/3.0/).

in HASMG G369a. Elsewhere, HASMG G369a shares with spinosaurids a lack of interdenticular sulci (*Hendrickx et al., 2019*).

The enamel texture of HASMG G369a is unusual in that two morphotypes are present: a veined textured basally and a more irregular texture apically. The former is common in

**Table 5 Results of the cluster analyses on the various iterations of the pan-theropodan datasets, with HASMG G369a treated as an unknown taxon.**

| Dataset | Cluster analysis | |
|---|---|---|
| | **Hierarchical clustering** | **Neighbour joining** |
| Whole dataset | *Suchomimus* | *Suchomimus* |
| Whole dataset (no denticles = ?) | *Suchomimus* | *Suchomimus+Irritator* |
| Reduced dataset | *Suchomimus* | *Suchomimus* |
| Reduced dataset (no denticles = ?) | *Suchomimus* | *Suchomimus* |

spinosaurids and synapomorphic for the clade: it is present in *Baryonyx* (NHMUK PV R9951), *Iberospinus* (ML 1190) and various cf. *Suchomimus* crowns (*e.g.*, MNN G35-9) (*Hendrickx et al., 2019*). Veined enamel texture is also present in *Ceratosuchops inferodios* (IWCMS 2014.95.5) and *Riparovenator milnerae* (IWCMS 2014.95.6) (C. Barker, pers. obs., 2022). Indeed, HASMG 639a also possesses the strong basal curvature of the veined texture towards the adjacent carinae, which is characteristic of the clade (*Hendrickx et al., 2019*; *Mateus et al., 2011*). However, an irregular enamel texture has so far only been reported for some *Irritator* crowns among spinosaurids (*Hendrickx et al., 2019*).

Differences in dental characters have been used to discuss the taxonomy of isolated spinosaurid teeth (*Fanti et al., 2014*; *Richter, Mudroch & Buckley, 2013*), though the utility of several traits has been questioned (*Hendrickx, Mateus & Buffetaut, 2016*). Tooth-bearing spinosaurid bones often lack erupted in-situ teeth, rendering variation between teeth within a complete tooth row poorly understood. Where teeth can be assigned to a single individual, as in the *Baryonyx walkeri* holotype NHMUK PV R9951, variation in ornamentation is documented (*Hendrickx, Mateus & Buffetaut, 2016*). Theropod dentition is also known to vary ontogenetically (*Hendrickx et al., 2019*) and it remains possible that differences in spinosaurid crown ornamentation may reflect ontogeny or tooth position more than phylogenetic position (*Hendrickx, Mateus & Buffetaut, 2016*).

Spinosaurid teeth are sometimes confused for those of crocodyliforms (*Bertin, 2010*; *Buffetaut, 2010*; *Hone, Xu & Wang, 2010*; *Sánchez-Hernández, Benton & Naish, 2007*), and the latter are well represented and taxonomically diverse in the Purbeck Group and Wealden Supergroup of southern England (*Benton & Spencer, 1995*; *Salisbury, 2002*; *Salisbury & Naish, 2011*). The crocodyliform fauna recovered from the Hastings Group is dominated by goniopholidids but also includes atoposaurids, bernissartiids and indeterminate mesoeucrocodylians and eusuchians (*Salisbury & Naish, 2011*). However, we can confidently dismiss a crocodyliform origin for HASMG G396a based on several lines of evidence.

Numerous "ridges" (*i.e.*, flutes) ornament the enamel of goniopholidid and pholidosaurid crowns; in *Goniopholis* and *Pholidosaurus* for instance, these are well defined and closely packed (*Allain et al., 2022*; *De Andrade et al., 2011*; *Martin, Raslan-Loubatié & Mazin, 2016*; *Owen, 1840–1845*; *Owen, 1878*; *Owen, 1879*), whereas those of HASMG G369a are fewer and poorly defined. Interestingly, *Owen (1840–1845)* drew attention to the differences present between enamel ornamentation of "*Suchosaurus cultridens*" relative to that of *Goniopholis*. Smooth carinae are observed in goniopholidids generally,

although false-ziphodont serrations are present in some taxa (*e.g., G. kiplingi*) (*De Andrade et al., 2011*; *Puértolas-Pascual, Canudo & Rabal-Garcés, 2015*; *Salisbury et al., 1999*). The latter are clearly distinguishable from the true denticles of HASMG G369a. Similarly, the mesial and distal carinae of pholidosaurids such as *Pholidosaurus* lack denticles, and can barely be differentiated from the flutes on the enamel surface (*Martin, Raslan-Loubatié & Mazin, 2016*). HASMG G396a is evidently not referable to atoposaurids, due both to the small size (<1m) of representative taxa (*e.g., Theriosuchus*) (*Schwarz & Salisbury, 2005*) and their distinctive distal dentition (*Salisbury & Naish, 2011*; *Young et al., 2016*). Fluted, conical teeth are present in the mesial dentition of bernissartiids, but these are also represented by small (<1m) taxa (*Martin et al., 2020*; *Sweetman, Pedreira-Segade & Vidovic, 2015*). In addition, their mesial teeth lack serrations and possess incipient cervical constriction (*Martin et al., 2020*; *Norell & Clark, 1990*). The short, rounded posterior crowns of bernissartiids are also obviously incompatible with the conidont morphology of HASMG G369a (*Martin et al., 2020*; *Norell & Clark, 1990*; *Sweetman, Pedreira-Segade & Vidovic, 2015*). In conclusion, we can reject with confidence the possibility that HASMG G369a might be considered referable to Crocodyliformes.

## The British spinosaurid record and biogeography of early spinosaurids

Most British spinosaurid skeletal (*i.e.,* non-dental) material has been recovered from the Barremian strata of Surrey (Upper Weald Clay Formation) and the Isle of Wight (Wessex Formation and base of the Vectis Formation) (*Barker et al., 2021*; *Barker et al., 2022*; *Charig & Milner, 1986*; *Charig & Milner, 1997*; *Martill & Hutt, 1996*; *Milner, 2003*). However, spinosaurid teeth are relatively common throughout the Wealden Supergroup (*Fowler, 2007*; *Turmine-Juhel et al., 2019*). While this is well known, the extent of the British spinosaurid record, and how it compares to that of other localities globally, has yet to be rigorously analysed.

The spinosaurid crown BEXHM 1995.485 is briefly described by *Charig & Milner (1997)* as originating from the "Ashdown Sand (Hauterivian)" near Bexhill in East Sussex, which *Milner (2003)* considered to be the earliest record of Spinosauridae. The term "Ashdown Sands" is now defunct (*Hopson, Wilkinson & Woods, 2008*), having been introduced by *Drew (1861)* before being formalised to Ashdown Formation by *Rawson (1992)*. The latter is now considered late Berriasian to early Valanginian in age (*Hopson, Wilkinson & Woods, 2008*). More recently, *Turmine-Juhel et al. (2019)* described and figured two poorly preserved crowns (BEXHM 2019.49.251 and BEXHM 2019.49.253) which they referred to *Baryonyx* sp. All three teeth were found at the same site—the Pevensey Pit at Ashdown Brickworks (Turkey Road, Bexhill-on-Sea; J Porter, D Brockhurst, pers. comm., 2022)—where the only exposures are of the Valanginian Wadhurst Clay Formation (*Turmine-Juhel et al., 2019*). BEXHM 1995.485 therefore cannot be Hauterivian or from the Ashdown Formation, contra *Charig & Milner (1997)* and *Milner (2003)*.

Modern interest in spinosaurids has resulted in the discovery of several Wealden Supergroup teeth in collections of crocodylomorph material housed in various institutions (*Buffetaut, 2007*; *Buffetaut, 2010*; *Fowler, 2007*; *Milner, 2003*). However, the historic nature of many of these specimens impacts our ability to identify their precise stratigraphic

position. *Fowler (2007)* described a pair of spinosaurid crowns within a collection of goniopholidid teeth (NHMUK PV R1901) from the "Wealden" of Hastings, a provenance which would make them Valanginian or possibly Berriasian. Elsewhere, *Bertin (2010)*, following *Lydekker (1888)*, listed a "*Suchosaurus cultridens*" crown (NHMUK PV R635) as originating from the Berriasian-Valanginian "Hastings Sands" of Sandown. Older works suggested that the "Hastings Sands" were represented on the Isle of Wight (*White, 1921*). However, the oldest exposed Wealden Supergroup strata on the Isle of Wight are from the entirely Barremian upper portion of the Wessex Formation (*Radley & Allen, 2012b*; *Sweetman, 2011*) and this specimen is probably Barremian in age.

It would thus appear that the oldest British spinosaurid material is definitively Valanginian in age, with Berriasian occurrences remaining a possibility for some specimens of undetermined provenance. In comparison, the oldest specimens from Iberia—the other European hotspot for spinosaurid remains—are late Hauterivian in age (*Malafaia et al., 2020*). *Fowler (2007)* described and figured a "saurian" tooth (DCM-G95a) potentially recovered from the Purbeck Group of Swanage (Dorset, UK), which possesses several spinosaurid characters such as fluted enamel ornamentation. However, it is not dissimilar to plesiosaur tooth crowns (*Fowler, 2007*) and is indeed most likely from a marine reptile (D Fowler, pers. comm., 2022).

Alleged Jurassic spinosaurid teeth have been reported from Tanzania (*Buffetaut, 2012*) and Niger (*Serrano-Martínez et al., 2015*; *Serrano-Martínez et al., 2016*). However, similarities with other theropod clades (notably ceratosaurs and megalosaurids) have been noted and doubts have been cast on the identification of these specimens (*Hendrickx et al., 2019*; *Soto, Toriño & Perea, 2020*). An additional putative spinosaurid tooth—initially compared with the above mentioned Tanzanian material—has been described from the Jurassic of France (*Vullo et al., 2014*). Insufficient data exists to regard this identity as secured and, like the above Tanzanian "spinosaurid" specimens, it is probable that this tooth is also non-spinosaurid. Thus, whilst Spinosauridae likely evolved during the Jurassic (*Barker et al., 2021*; *Carrano, Benson & Sampson, 2012*), definitive Jurassic material pertaining to the group remains elusive. Moreover, associated discussion regarding the early evolution of spinosaurid teeth, with a proposed gradual acquisition of adaptations towards piscivory (*Buffetaut, 2012*; *Serrano-Martínez et al., 2015*; *Serrano-Martínez et al., 2016*), are best considered speculative pending further data (*Hendrickx et al., 2019*; *Soto, Toriño & Perea, 2020*).

A small, conidont crown (LPUFS 5737) from the Berriasian–Valanginian of Brazil (*Sales et al., 2017*) may represent one of the oldest spinosaurid occurrences globally. Additional spinosaurine teeth, as well as specimens referred to Baryonychinae (*e.g.*, LPUFS 5870) or regarded as indeterminate spinosaurids (*e.g.*, LPUFS 5871), have also been recently recovered from the locality (*Aragão, 2021*; *Lacerda et al., 2023*). We note that the identification of these specimens is based on (sometimes limited) qualitative data and would benefit from additional support generated using cladistic, discriminant and cluster analyses, as advocated for isolated theropod teeth in general (*Hendrickx, Tschopp & Ezcurra, 2020*). Nevertheless, evidence for spinosaurids in deposits of Berriasian–Valanginian age could complicate the biogeographic scenario proposed for the clade by *Barker et al. (2021)*,

as independently suggested by *Lacerda et al. (2023)*. *Barker et al. (2021)* regarded Europe as the ancestral region but did not include specimens known from isolated teeth in their analyses. As a result, alternative biogeographical scenarios include earlier instances of dispersal from the proposed European ancestral area, or a different ancestral area altogether.

## Spinosaurid persistence in the Late Cretaceous and status of specimen XMDFEC V10010

The results of the discriminant function analyses (Supplementary Information) show that XMDFEC V10010 does not associate with Spinosauridae when classified at either the clade or genus level. At the clade-level, the specimen was consistently classified as an allosauroid (Metriacanthosauridae or Allosauridae; reclassification rates = 54.46–62.12%; PC1 37.97–57.88%, PC2 19.11–31.01%), regardless of the dataset or whether serrations were considered inapplicable. At the genus-level, the allosauroid signal was retained, with the tooth most commonly referred to Early Cretaceous *Erectopus*, a tetanuran previously referred to Allosauroidea (and possibly Metriacanthosauridae) (*Carrano, Benson & Sampson, 2012*). XMDFEC V10010 was also referred to the megalosauroid *Condorraptor* and the abelisaurid *Skorpiovenator* in some genus-level DFAs. Reclassification rates in the genus level analyses were generally similar to those at the clade level analyses, and ranged between 57.4–63.68%.

The cluster analyses using the hierarchical clustering option consistently recovered XMDFEC V10010 as the sister taxon to an indeterminate abelisaurid. Similarly, the neighbour-joining option also commonly recovered the tooth as sister to an indeterminate abelisaurid, with several analyses of the whole dataset also recovering XMDFEC V10010 as a sister taxon to Abelisauridae indet. + *Fukuiraptor*.

The conflicting signals produced by the above quantitative analyses on XMDFEC V10010 are perhaps expected given that the dentition of Metriacanthosauridae and Allosauridae are considered the closest to that of Abelisauridae (*Hendrickx, Tschopp & Ezcurra, 2020*), although these allosauroid clades are not known from the Late Cretaceous (*Carrano, Benson & Sampson, 2012*). In comparison, abelisaurids were successful and diverse during the Late Cretaceous but are poorly represented in Asian deposits (outside of India) (*Carrano & Sampson, 2008*; *Delcourt, 2018*). Their teeth are nevertheless relatively diagnostic; however, the dental characters that unite Abelisauridae involve the shape of the premaxillary and maxillary alveoli (which are unknown for XMDFEC V10010) or relate to the morphology of the denticles (which are somewhat worn in XMDFEC V10010; *Hone, Xu & Wang, 2010*) (*Hendrickx et al., 2019*; *Hendrickx, Tschopp & Ezcurra, 2020*). Cladistic analyses of XMDFEC V10010 based on first hand examination of the specimen would be beneficial, and we refrain from referring the tooth to a theropod clade without this additional line of evidence. However the quantitative evidence presented herein corroborates previous suggestions that XMDFEC V10010 cannot be referred to Spinosauridae (*Buffetaut et al., 2019*; *Katsuhiro & Yoshikazu, 2017*; *Soto, Toriño & Perea, 2020*). With the Patagonian late Cenomanian-early Turonian tooth referred to Spinosauridae in *Salgado et al. (2009)* also likely from a different theropod lineage (*Soto, Toriño & Perea, 2020*), the youngest definitive

spinosaurid remains appear to come from Cenomanian deposits of Africa (*Benyoucef et al., 2022*; *Ibrahim et al., 2020b*; *Sereno et al., 2022*).

Assuming the reinterpretation of the above-mentioned Chinese and Patagonian specimens is correct, the potential extinction of Spinosauridae around the Cenomanian–Turonian boundary (CTB) remains poorly understood (*Candeiro, Brusatte & De Souza, 2017*). This time interval coincides with the peak Cretaceous greenhouse climate and a major marine transgression, and a marine extinction event has been documented (*Kerr, 2014*; *Sepkoski, 1986*). However, studies of the faunal changes in terrestrial, freshwater and brackish water environments during this transition are rare, and available data from North America suggests these faunas were not (a few taxa excepting) overly affected (*Benson et al., 2013*; *Eaton et al., 1997*). Spinosaurids are not definitively known from the Mesozoic of North America, however, and it may be that results inferred from these deposits may not be applicable elsewhere. Moreover, as theropods that have been positively associated with costal palaeoenvironments (*Sales et al., 2016*), it is interesting to speculate upon the impact of the CTB marine transgression on available spinosaurid habitat, and certainly warrants further consideration as a potential driver of their apparent extinction.

## CONCLUSIONS

An isolated spinosaurid tooth crown HASMG G369a cannot be referred to *Baryonyx* based on the results of multiple quantitative and qualitative analyses, and further supports suggestions that multiple spinosaurid taxa are present within the Wealden Supergroup. Although the precise provenance of HASMG G369a could not be ascertained with certainty, it is among the oldest spinosaurid remains found in Britain and is probably Valanginian in age. Indeed, while the oldest definitive British spinosaurid material comes from this stage, Berriasian occurrences cannot be completely ruled out for some specimens. Future work should look to apply cladistic and discriminant methods on spinosaurid crowns from known strata within the Wealden Supergroup, which may help further assess the British diversity of the clade and provide information on the dental evolution of these atypical theropods.

Following the general consensus that Jurassic spinosaurid material is currently unknown, and that previously referred material represent other theropod clades (see above), a literal interpretation of the fossil record highlights Western Europe as a key region for early spinosaurid evolution, given the wealth of (albeit largely fragmentary) Early Cretaceous material. However, the presence of isolated spinosaurid teeth from the Berriasian-Valanginian of Brazil suggests that early spinosaurids were more spatially widespread, and underlines the palaeobiogeographical importance of fragmentary specimens. As such, alternative biogeographic scenarios regarding the place of origin and early movements of the clade should be examined. Meanwhile, evidence for post-Cenomanian spinosaurid persistence is not supported based on quantitative reinterpretation of dental material previously referred to the clade, and the lack of spinosaurid remains in the latter stages of the Cretaceous hints at an extinction event around the Cenomanian-Turonian boundary.

**Institutional Abbreviations**

| | |
|---|---|
| **BEXHM** | Bexhill Museum, Bexhill, UK |
| **DCM** | Dorset County Museum, Dorchester, UK |
| **HASMG** | Hastings Museum and Art Gallery, Hastings, UK |
| **IWCMS** | Dinosaur Isle Museum (Isle of Wight County Museum Services) Sandown, Isle of Wight (UK) |
| **IVPP** | Institute of Vertebrate Paleontology and Paleoanthropology, Beijing, China |
| **LUPFS** | Laboratorio de Paleontologia of the Universidade Federal de Sergipe, São Cristóvao, Brazil |
| **ML** | Museu de Lourinhã, Lourinhã, Portugal |
| **MNEMG** | Maidstone Museum, Kent, UK |
| **MNN** | Musée National du Niger, Niamey, Republic of Niger |
| **NHMUK** | Natural History Museum, London, UK |
| **SMNS** | Staatliches Museum für Naturkunde Stuttgart, Germany |
| **XMDFEC** | Xixia Museum of Dinosaur Fossil Eggs of China, Xixia, China |

# ACKNOWLEDGEMENTS

We would like to thank: Phil Hadland for access to the specimen and discussions regarding provenance; Julian Porter and David Brockhurst for information pertaining to BEXHM 1995.485; Christophe Hendrickx for discussion regarding methods; Denver Fowler for discussion regarding spinosaurid dentition; Luis Coy and Charlotte Collier for help with imagery; and Andrea Cau for help with the programme TNT. Martin Simpson is additionally thanked for assisting DN's research on the historic specimens accessioned at HASMG and BEXMH. The programme TNT is made available thanks to the Willi Hennig Society. Thanks are also extended to editor Fabien Knoll and reviewers Kirstin Brink and Marco Sales for their attention to this work.

## Funding

Chris Barker was funded by the Institute for Life Sciences (IFLS; University of Southampton) and the Engineering and Physical Sciences Research Council (EPSRC; No. 2283360). The funders had no role in study design, data collection and analysis, decision to publish, or preparation of the manuscript.

## Grant Disclosures

The following grant information was disclosed by the authors:
Institute for Life Sciences (IFLS; University of Southampton).
Engineering and Physical Sciences Research Council: EPSRC; No. 2283360.

## Competing Interests

The authors declare there are no competing interests.

## Author Contributions

- Chris T. Barker conceived and designed the experiments, performed the experiments, analyzed the data, prepared figures and/or tables, authored or reviewed drafts of the article, and approved the final draft.
- Darren Naish conceived and designed the experiments, analyzed the data, authored or reviewed drafts of the article, and approved the final draft.
- Neil J. Gostling conceived and designed the experiments, analyzed the data, authored or reviewed drafts of the article, and approved the final draft.

## Data Availability

Raw data is available in the Supplemental Files.

The supplementary files include the character matrix (.tnt) for the phylogenetic analysis of HASMG G369a, as well as morphometric data files (.dat) for the DFA and cluster analyses of HASMG G369a and XMDFEC V10010.

## Supplemental Information

Supplemental information for this article can be found online at http://dx.doi.org/10.7717/peerj.15453#supplemental-information.

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
