# Peer review of "Isolated tooth reveals hidden spinosaurid dinosaur diversity in the British Wealden Supergroup (Lower Cretaceous)"

_PeerJ, doi:10.7717/peerj.15453_

## Round 0.1 · original submission · Minor Revisions

I note that both reviewers are very positive. However, Reviewer 2 has made a number of suggestions that I would like you to consider.

·

Basic reporting

This is a well-written, highly detailed paper describing and identifying a spinosaurid tooth. The references are encompassing. The figures are nice and clearly show tooth detail and results of statistical analyses.

Experimental design

The experimental design is appropriate for this study. Methods are described in extremely sufficient detail in order to be repeated, and necessary data are provided.

Validity of the findings

Interpretations of the results are well supported and conclusions are well stated. This study sets up interesting questions for future research projects around spinosaurid diversity.

Additional comments

This is a nice, rigorous paper that adds to our overall knowledge of dinosaur diversity and paleobiogeography of spinosaurids.

·

Basic reporting

The text is very well written and structured, with all relevant references included. Thus, I believe the main issue regarding the organization of the paper is that there are two Tables 1. So it is necessary to rename the 'second' Table 1 as Table 2 and do the same for the following tables. Maybe you can delete the last section of your discussion, as it is not among the main goals of your work, but I will leave it up to you and the editor if you all agree that is better to make the text shorter.

Experimental design

The methodology follows the most recent and often adopted protocols available in the literature and it is explained with all relevant details.

Validity of the findings

As the methodology seems perfectly appropriate to me, the results also sound reasonable and convincing. The conclusions are summarized according to the main goals of the paper. However, with respect to the last section of the Discussion (Spinosaurid persistence in the Late Cretaceous and status of specimen XMDFEC V10010), I am not sure if it is in fact relevant or necessary regarding original goals of your work. I mean it especially due to the still speculative nature of such a discussion in view of the present lack of enough data for better supporting it.

Additional comments

I have made some comments throughout the text pointing some minor issues related to possible mistakes regarding the writing of the text as a whole (including figure legends). Please find them in the annotated PDF file of your manuscript that I have uploaded during the submission of my review. I have also commented on some statements, focusing on scientific issues, most of which reflects more my particular view on the subject than something that is mandatory to you to agree and follow. Thus, it is up to you to decide whether or not to follow my suggestions, unless the editor believes you should do so.

---

## Round 0.2 · accepted · Accept

Please, insert “UK” after “Isle of Wight” line 844.